# Past political violence and interpersonal violence against children and youth in Africa

Marcella Vigneri[1,2], Olusegun Fadare [3], Karen Devries [1], Vegard Iversen [3] & Tilman Brück [3,4,5,6] ✉

Political violence (the use of force by groups for political motives) adversely affects children and youth, but there is limited rigorous evidence on whether exposure to political violence is associated with a higher risk of subsequent interpersonal violence against adolescents and young adults. We estimate this association with multi-country micro data from nine African countries, merging nationally representative Violence Against Children and Youth Survey (VACS) data with Armed Conflict Location and Event Data (ACLED) for Côte d'Ivoire, Kenya, Malawi, Mozambique, Namibia, Nigeria, Uganda, Zambia, and Zimbabwe. We find that a one standard deviation increase in past long-term exposure to political violence increases the odds of adolescents and young adults subsequently experiencing emotional violence from family members by 5.5%, physical violence from intimate partners by 3.0%, and sexual violence in poorer households by 16.9%. There was no statistically significant association for past short-term exposure to political violence.

In 2022, about one in six children – or close to 470 million children globally – lived in areas affected by violent conflict[1]. The immediate effects of large-scale armed conflict on children, adolescents, and young adults, including death, disease, and displacement, have been extensively researched[2,3]. Comparatively less is known about the effects of short- or longer- term exposure to armed conflict or other forms of political violence (the use of force by a group with a political motive) on the risk of children, adolescents, and young adults experiencing subsequent interpersonal violence. These knowledge gaps stem both from a lack of data and from research often being conducted in disciplinary silos[4].

In addition to children's direct exposure to violence from conflict, proximity to violent conflict may trigger adverse stress and other psychological reactions, including post-traumatic stress disorder, in children themselves, in children's caregivers, within communities and in society more broadly[5–7]. Children, adolescents, and young adults in areas affected by violent conflict often experience violence from those responsible for their care[8]. Evidence from conflict settings shows that adolescent girls and women who witness community violence, or conflict violence, are more likely to report childhood violence from caregivers[9]. It is also more common for women to experience violence from intimate partners and become more accepting of intimate partner violence (IPV)[4,10,11]. Intimate partner violence against women in conflict settings has knock-on effects on parenting: women experiencing IPV are more likely to suffer from mental health problems, to resort to harsher parenting[12], and to have adolescent children who experience more stigma in their communities[13]. Similarly, parents in war and conflict settings use more corporal punishment and other harsh parenting techniques[6,14,15]. In Northern Uganda, women's re-victimisation experiences and men's psychopathological symptoms contribute to intergenerational transmission of violent behaviours[6].

Further, children's exposure to violent contexts has several well-documented adverse mental health and developmental consequences, including increased risk of symptoms of depression and anxiety and externalising disorders such as hyperactivity and conduct disorders[16]. Understandably, the prevalence of children with symptoms of, or clinically diagnosable, post-traumatic stress disorder is much higher in settings affected by violent conflict[17–19]. Symptoms of common mental disorders are associated with incident experiences of intimate partner

[1]London School of Hygiene and Tropical Medicine, London, UK. [2]Save the Children UK, London, UK. [3]Natural Resources Institute, University of Greenwich, Chatham Maritime, UK. [4]Humboldt-Universität zu Berlin, Thaer-Institute, 10099 Berlin, Germany. [5]Leibniz Institute of Vegetable and Ornamental Crops, Großbeeren, Germany. [6]ISDC – International Security and Development Center, Berlin, Germany. ✉e-mail: tilman.brueck@hu-berlin.de

violence in women[20], underscoring the bidirectional nature of the relationship between poor mental health and violence.

For children, adolescents and young adults exposed to conflict, poor mental health is also likely to affect their subsequent risk of violence victimisation. This could be for a variety of reasons. Speculatively, perpetrators of violence often seek out victims whom they perceive to be vulnerable, and children struggling with depression may be visibly anhedonic, lethargic, and withdrawn, making them easy targets. For children with externalising symptoms, including aggressive behaviour, caregivers and others may not recognise this as an expression of distress and may punish children more harshly, especially in contexts where physical discipline is the norm.

There are additional pathways by which living in areas affected by violent conflict may result in increased risk of interpersonal violence against children, adolescents, and young adults. Political violence may also set off other forms of violence by raising tensions between communities and social groups[21]. During war time, people may learn to use violence as a coping strategy in interpersonal relations. Soldiers exposed to wartime violence against women are more likely to be violent intimate partners long after the war ends, pointing to possible longer-term alterations to behaviour in relationships[22]. Boys observing fathers commit domestic violence are more prone to becoming domestic abusers as adults[23].

The risk of violence against children, adolescents, and young adults may be further exacerbated by political violence-induced weakening or breakdown of traditional, community, and other support systems, public health and other infrastructure or other child protection systems and institutions. For example, children and adolescents in areas affected by armed conflict may not be able to attend school. This means that there may be fewer opportunities for children and adolescents to report violence, or for adults who are not their caregivers to detect it; it also deprives children and young adults of education about sex and healthy relationships[24] which is their right under the United Nations Convention on the Rights of the Child, and of skills to resolve interpersonal conflicts.

Similarly, official child protection structures may not be functioning. While children and adolescents seldom use official reporting structures to disclose violence against them[25], the absence of such options may prevent children and adolescents who have experienced violence from obtaining support to tackle resulting poor mental health, which may heighten the risk of future violence.

This paper presents the results from a study of the association between past political violence (PolV) and subsequent interpersonal violence against children, adolescents, and young adults (abbreviated VAC in the spirit of the data sets we analyse, which record self-reported experiences of physical, sexual, and emotional violence by children, adolescents, and young adults). In our analysis, we draw on a cross-country nationally representative micro-level dataset that we created. This dataset combines temporally and spatially disaggregated information on violence against children, adolescents, and young adults of wide coverage and depth from surveys conducted across different years, with disaggregated data on political violence. To construct this dataset, we merge data from Violence Against Children and Youth Surveys (VACS) with Armed Conflict Location and Event Data (ACLED) for nine African countries, namely Côte d'Ivoire, Kenya, Malawi, Mozambique, Namibia, Nigeria, Uganda, Zambia, and Zimbabwe. We first examine whether past short-term or long-term exposure to political violence increases the likelihood of children, adolescents and young adults experiencing recent physical, sexual or emotional violence. Next, we consider whether such an adverse role of PolV exposure varies across PolV categories. Finally, we examine whether the likelihood of VAC varies with age, gender and household poverty status. Following the main analysis, we undertake a series of robustness tests.

In this study, we find that the average prevalence of past-year physical, sexual and emotional violence against adolescents and young adults in our merged dataset across nine African countries is 23.1%, 10.3%, and 11.2%, respectively. A one standard deviation increase in past long-term exposure to political violence increases the odds of adolescents and young adults subsequently experiencing emotional violence from family members by 5.5% ($p < 0.001$), physical violence from intimate partner by 3.0% ($p = 0.047$), any violence by 3.0% ($p = 0.013$), and at least two types of violence by 4.3% ($p = 0.006$). We also find that adolescents and young adults from poorer (lower wealth quintile) households are 16.9% ($p < 0.001$) more likely to experience sexual violence. There was no statistically significant association for past short-term exposure to political violence. Our findings highlight the importance of considering political and historical contexts in efforts to mitigate violence against adolescents and young adults within families, in schools, and in early intimate partnerships. Our results open avenues for future research and programming to achieve SDG 16.2, the prevention of violence against children and youth.

## Results

### Descriptive statistics

Using subnational aggregates of political violence events for the 15 years preceding each survey, we analyse interpersonal violence data from nine African countries, comprising 35,439 respondents: 15,888 adolescents aged 13-17 years (44.8% of the sample) and 19,548 young adults aged 18-24 years (55.2%) (Table 1). Female respondents are 66.7% of the sample due to an oversampling of girls; respondents self-identified their gender. The average respondent was 18 years old; 49% were attending school at the time of the survey and 24% had worked during the past 12 months.

In the year preceding each country survey, 23.1% of respondents reported experiencing physical violence (PV), 10.3% experienced sexual violence (SV), and 11.2% experienced emotional violence (EV). 32.6% of respondents reported experiencing any violence (AV) while 10.0% experienced multiple types of violence (MV).

The average number of political violence events per 100,000 population in an administrative area was 27.0, comprising 5.6 battles, 0.4 explosions, 16.8 events with violence against civilians, 3.2 riots, and 1.1 strategic development events. Given that we weigh the political violence events by the population of each administrative area, these values differ from the absolute number of political violence events (Fig. 1). For example, Namibia recorded few political violence events in total and has a small population, resulting in a high rate of political violence events per 100,000 population, whereas Nigeria recorded many political violence events in total and has a large population, resulting in a low rate of political violence events per 100,000 population.

### Regression results

Figure 2 plots the adjusted odds ratios (AOR) and their confidence intervals (with full results in Table A3). Our cross-country analysis finds no statistically significant associations between short-term (one to five years) exposure to PolV and VAC, irrespective of the type of VAC experienced (Fig. 2). However, long-term exposure (over a 15-year time-period) to PolV increases the odds of experiencing VAC. Specifically, a one standard deviation increase in political violence is associated with a 5.5% increase in the odds of experiencing emotional violence (AOR 1.055; $p < 0.001$). We also find that the odds of adolescents and young adults experiencing AV (AOR 1.030; $p = 0.013$) or MV (AOR 1.043; $p = 0.006$) are statistically significant at the 5% and 1% levels, respectively.

Figure 3 shows how long-term PolV, disaggregated by event categories, is correlated with VAC. We find that the odds of experiencing EV increase for all PolV event categories, with effect sizes

## Table 1 | Descriptive statistics

| Variable | Measurement | Mean | SD | Min | Max |
|---|---|---|---|---|---|
| **Individual and household characteristics** | | | | | |
| Age in years | Age of respondents (year) | 18.15 | 3.44 | 13 | 24 |
| Age category | 1=aged 18-24; 0=aged 13-17 | 0.55 | 0.50 | 0 | 1 |
| Gender | 1=Female; 0= Male | 0.67 | 0.47 | 0 | 1 |
| Marital status | 1=Respondent presently lives with spouse | 0.24 | 0.43 | 0 | 1 |
| School attendance | 1=Respondent is currently attending school | 0.49 | 0.50 | 0 | 1 |
| Recently worked | 1=Respondent worked during the past 12 months | 0.24 | 0.43 | 0 | 1 |
| Wealth index | Household wealth index | 0.35 | 0.26 | 0 | 1 |
| Wealth quintile indicator | 1=Low-Lower-Middle; 0=Higher-Highest wealth quintiles | 0.60 | 0.49 | 0 | 1 |
| **Past year VAC** | | | | | |
| PV | Experienced Physical Violence (PV) by any perpetrator | 0.23 | 0.42 | 0 | 1 |
| SV | Experienced Sexual Violence (SV) by any perpetrator | 0.10 | 0.30 | 0 | 1 |
| EV | Experienced Emotional Violence (EV) by adult family member | 0.11 | 0.32 | 0 | 1 |
| AV | Experienced Any Violence (AV) type | 0.33 | 0.47 | 0 | 1 |
| MV | Experienced Multiple Violence (MV), i.e., two or more violence types | 0.10 | 0.30 | 0 | 1 |
| PV-family | Experienced PV by adult family member | 0.08 | 0.28 | 0 | 1 |
| PV-partner [a] | Experienced PV by intimate partner | 0.07 | 0.26 | 0 | 1 |
| PV-peer | Experienced PV by peer | 0.09 | 0.29 | 0 | 1 |
| PV-comm | Experienced PV by adult in the community | 0.10 | 0.30 | 0 | 1 |
| **Long-term political violence (total per admin division)** | | | | | |
| Political violence | Number of political violence events over 15 years before VACS | 135.93 | 396.16 | 0 | 2482 |
| **Long-term political violence (Over 15-year time-period, measured per 100,000 population in admin division)** | | | | | |
| Political violence | Political violence events (number of) | 27.02 | 67.30 | 0 | 903.09 |
| Battles | Battles events (number of) | 5.56 | 32.05 | 0 | 507.36 |
| Explosions | Explosions/remote violence events (number of) | 0.43 | 3.95 | 0 | 71.03 |
| Violence against civilians | Violence against civilians events (number of) | 16.76 | 37.46 | 0 | 526.11 |
| Riots | Riot/mob violence events (number of) | 3.20 | 6.93 | 0 | 101.47 |
| Strategic developments | Strategic developments (number of) | 1.07 | 2.57 | 0 | 30.44 |
| Protests | Protests (number of) | 5.82 | 11.83 | 0 | 51.44 |

The number of observations is 35,439. We transformed the political violence indicators to standard deviations to facilitate spatial comparisons of political violence intensity. Data sources: Armed Conflict Location and Event Data (ACLED) and Violence Against Children and Youth Surveys (VACS). [a] Sample taken from individuals with partners ($n = 21,495$).

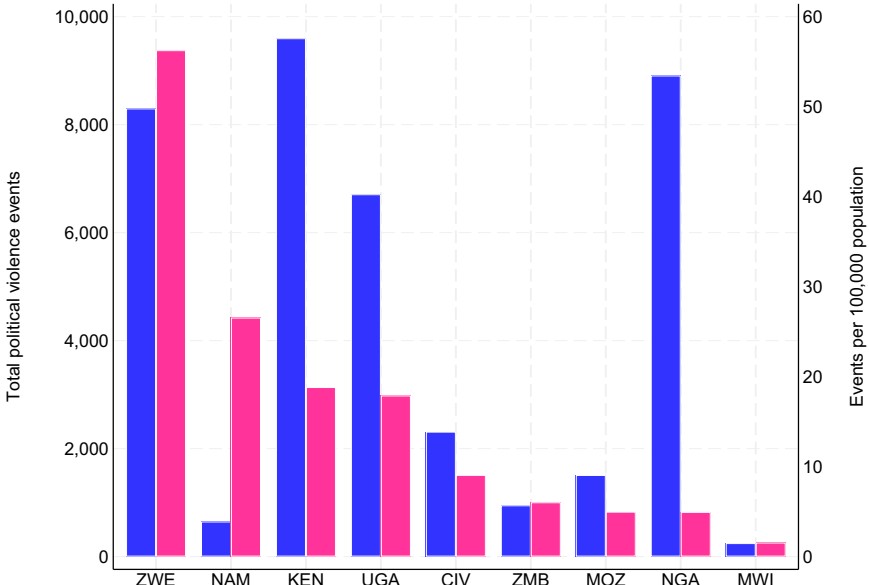

**Fig. 1 | Political violence events in nine African countries.** Total and population-adjusted political violence events in nine Violence Against Children and Youth Surveys (VACS) countries, 15-year window preceding each VACS reference year. Data sourced from Armed Conflict Location and Event Data (ACLED). Blue bars on the left (left axis) indicate the total number of ACLED-recorded political violence events aggregated over the 15 years before each country's VACS reference year. Red bars on the right (right axis) indicate the corresponding rate, expressed as events per 100,000 population per country. Countries are ordered by the relative number of political violence events.

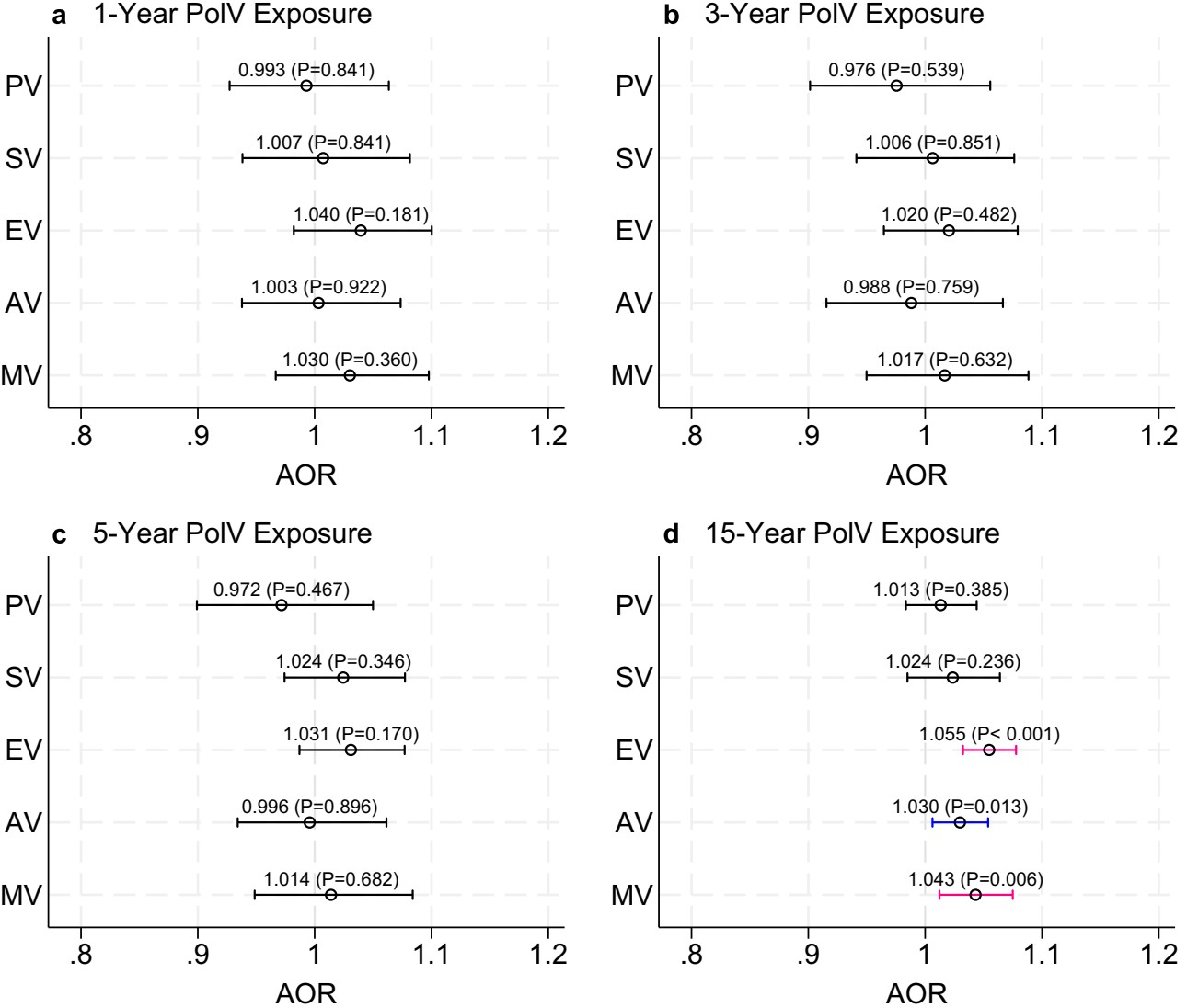

**Fig. 2 | Association between exposure to political violence and violence against children and youth by exposure duration.** Adjusted odds ratios (AOR) for exposure to political violence (PolV) over 1 year (**a**), 3 years (**b**), 5 years (**c**), and 15 years (**d**). PV: Physical Violence; SV: Sexual Violence; EV: Emotional Violence; AV: Any Violence; MV: Multiple Violence. Aggregate political violence is defined as the number of political violence events in each administrative area over the specified period, standardised by population size and transformed into standard deviations. Models are adjusted for respondent age, education, employment, household wealth, partner status, and country fixed effects. Respondent sample size: $n = 35,439$. Blue and red bars indicate odds ratios that are statistically significant at $p < 0.05$ and $p < 0.01$, respectively. Robust standard errors clustered at the administrative division level. The bars indicate 95% confidence intervals of the estimated coefficient of interest (indicated by a small circle – the actual coefficient and its corresponding *P*-value are also listed). Full results for long-term (15-year) PolV exposure are available in Supplementary Table 3a–e: column 4.

ranging from a 2.3% increase for explosions (AOR 1.023; $p = 0.005$) to 10.3% for events with violence against civilians (AOR 1.104; $p < 0.001$). In contrast, the odds of experiencing SV increase only after riot events (AOR 1.038; $p = 0.001$). Battles and riots are correlated with several VAC outcomes. Notably, a one standard deviation increase in battle events increases the odds of EV by 3.8% (AOR 1.038; $p < 0.001$), AV by 2.1% (AOR 1.021; $p = 0.035$), or MV by 3.2% (AOR 1.033; $p = 0.002$). A similar rise in riot events raises the odds of experiencing EV by 4.0% (AOR 1.041; $p < 0.001$), AV by 3.7% (AOR 1.037; $p = 0.001$) or MV by 4.1% (AOR 1.041; $p < 0.001$).

Our analysis of heterogeneity shows that young adults are more likely to experience PV (AOR 1.092; $p = 0.024$) than adolescents (Fig. 4). Notably, there was no statistically significant difference by gender of adolescents and young adults of the odds of experiencing PV, SV or EV in the context of PolV. As PolV increases, respondents from poorer households are at a higher risk of experiencing SV (AOR 1.169; $p < 0.001$) than respondents from better-off households.

When disaggregating PV by perpetrator (Fig. 5), we find that PolV is associated with increased odds of experiencing PV from intimate partners (AOR 1.030; $p = 0.047$), but PolV is associated with decreased odds of experiencing PV from intimate partners for young adults compared to adolescents (AOR 0.938; $p = 0.031$). We also find increased odds of experiencing intimate partner violence (AOR 1.039; $p = 0.005$) for respondents from poorer households. Young adults also have higher odds of experiencing PV from family members than adolescents (AOR 1.157; $p < 0.001$).

### Robustness checks
Our main results remain robust across different model specifications (see Supplementary Tables 3a–e).

We first validate our results by replacing political violence with non-violent protests as the exposure indicator in a placebo test (see Supplementary Table 7). There was no statistically significant association between protests and any measure of VAC (all *P*-values > 0.10).

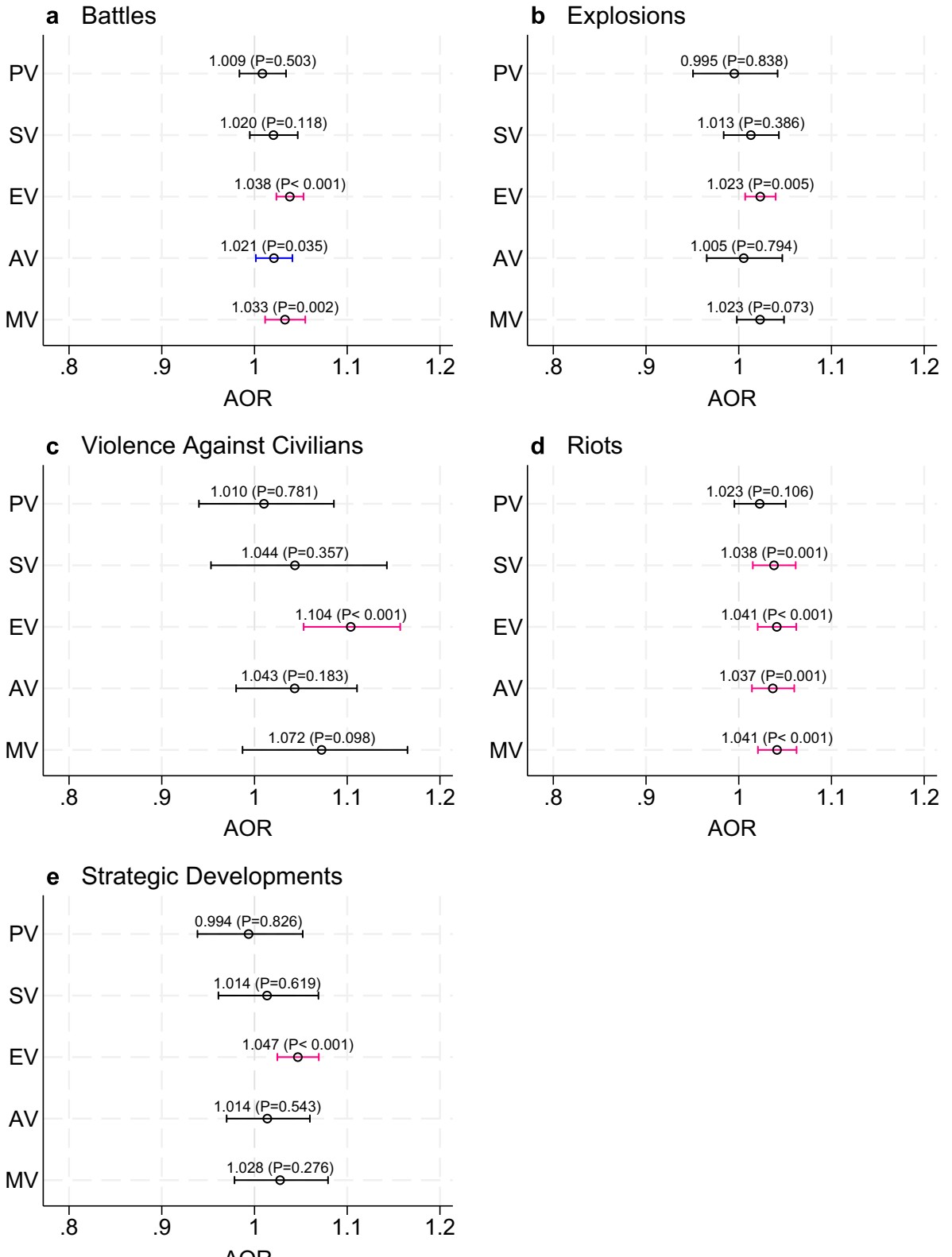

**Fig. 3 | Association between exposure to specific types of political violence and violence against children and youth.** Adjusted odds ratios (AOR) for exposure to battles (**a**), explosions (**b**), violence against civilians (**c**), riots (**d**), and strategic developments (**e**). PV: Physical Violence; SV: Sexual Violence; EV: Emotional Violence; AV: Any Violence; MV: Multiple Violence. Respondent sample size: $n = 35,439$. Blue and red bars indicate odds ratios that are statistically significant at $p < 0.05$ and $p < 0.01$, respectively. Robust standard errors clustered at the administrative division level. The bars indicate 95% confidence intervals of the estimated coefficient of interest (indicated by a small circle – the actual coefficient and its corresponding $P$-value are also listed). Models are adjusted for respondent age, education, employment, household wealth, partner status, and country fixed effects. Full results are available in Supplementary Table 4a–e.

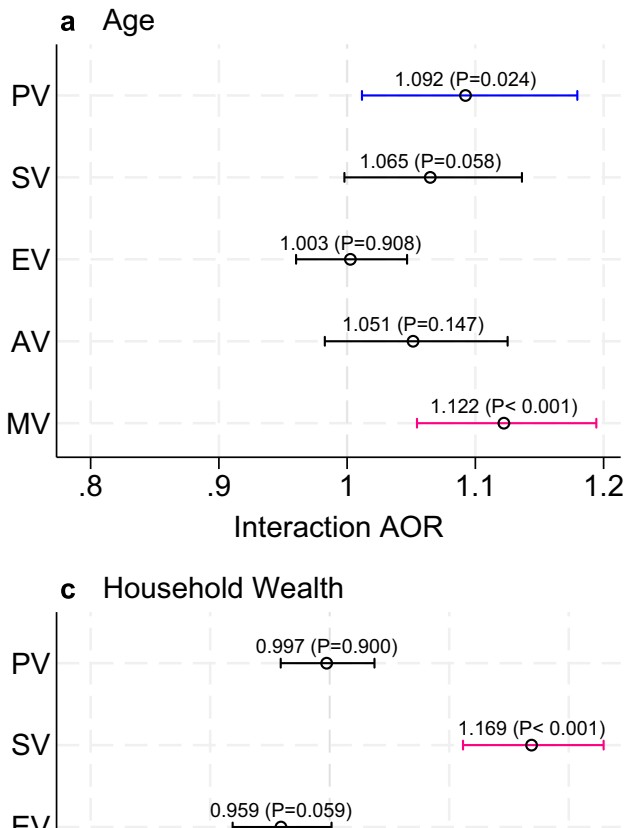

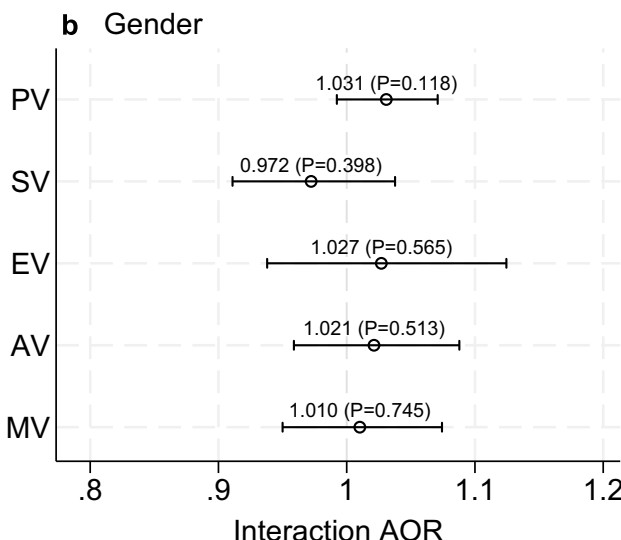

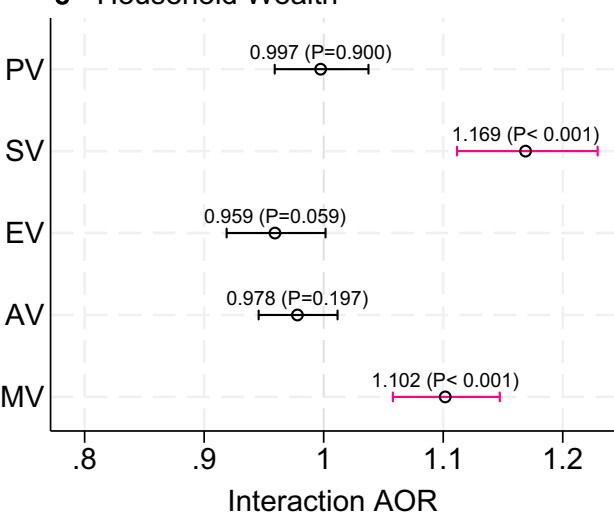

**Fig. 4 | Heterogeneity in the association between exposure to political violence and violence against children and youth.** Adjusted odds ratios (AOR) for the interaction terms between 15-year exposure to political violence (PolV) and respondent age (**a**): Youth (18–24 years) vs. children (13–17 years); gender (**b**): Female vs. male respondents; and household wealth (**c**): Lowest-low-middle vs. high-highest wealth quintiles. PV: Physical Violence; SV: Sexual Violence; EV: Emotional Violence; AV: Any Violence; MV: Multiple Violence. An AOR >1 indicates that the association between PolV and violence against children and youth is stronger for the specified subgroup (youth, females, poorest quintiles). Blue and red bars indicate odds ratios that are statistically significant at $p < 0.05$ and $p < 0.01$, respectively. Robust standard errors clustered at the administrative division level. The bars indicate 95% confidence intervals of the estimated coefficient of interest (indicated by a small circle – the actual coefficient and its corresponding *P*-value are also listed). Full interaction results are available in Supplementary Table 5a–c.

This suggests that the observed relationship between PolV and VAC is specific to exposure to political violence rather than to peaceful political protests.

Second, we estimate a model without individual, country-level or household-level controls, and then add these sequentially (see Supplementary Table 3a–e: columns 1, 2 and 4). The findings confirm that experiencing EV and MV remain significant ($p < 0.05$) with relatively stable effect sizes across specifications.

Third, we conduct a sensitivity test by re-estimating Eq. (1), excluding the relative household wealth (see Supplementary Table 3a–e: column 3). These models corroborate the strong and significant association between exposure to PolV and the odds of experiencing especially EV, with or without wealth included (AOR 1.054; $p < 0.001$ versus AOR 1.055; $p < 0.001$) (see Supplementary Table 3c: columns 3 and 4, respectively).

Fourth, when testing an alternative indicator of PolV (fatalities), the results are very similar to our previous analysis (see Supplementary Table A8a, b). A one standard deviation increase in fatalities over 15 years is associated with an increase in the odds of experiencing SV (AOR 1.023; $p = 0.021$), EV (AOR 1.033; $p < 0.001$) and MV (AOR 1.035; $p < 0.001$).

Finally, we re-estimate our main result using two non-overlapping windows of exposure to past political violence of 1–5 years and 6–15 years before the survey reference period, and include both in the same logistic model. Collinearity is modest, so the two coefficients are well identified. The estimates indicate that political violence exposure in the preceding 6–15 years is statistically significantly associated with an increased EV only (see Supplementary Table A9). This confirms that it is past political violence from more than five years ago that is associated with emotional violence against adolescents and young adults today.

## Discussion

While short-term exposure to political violence is not statistically significantly associated with any form of violence against children,

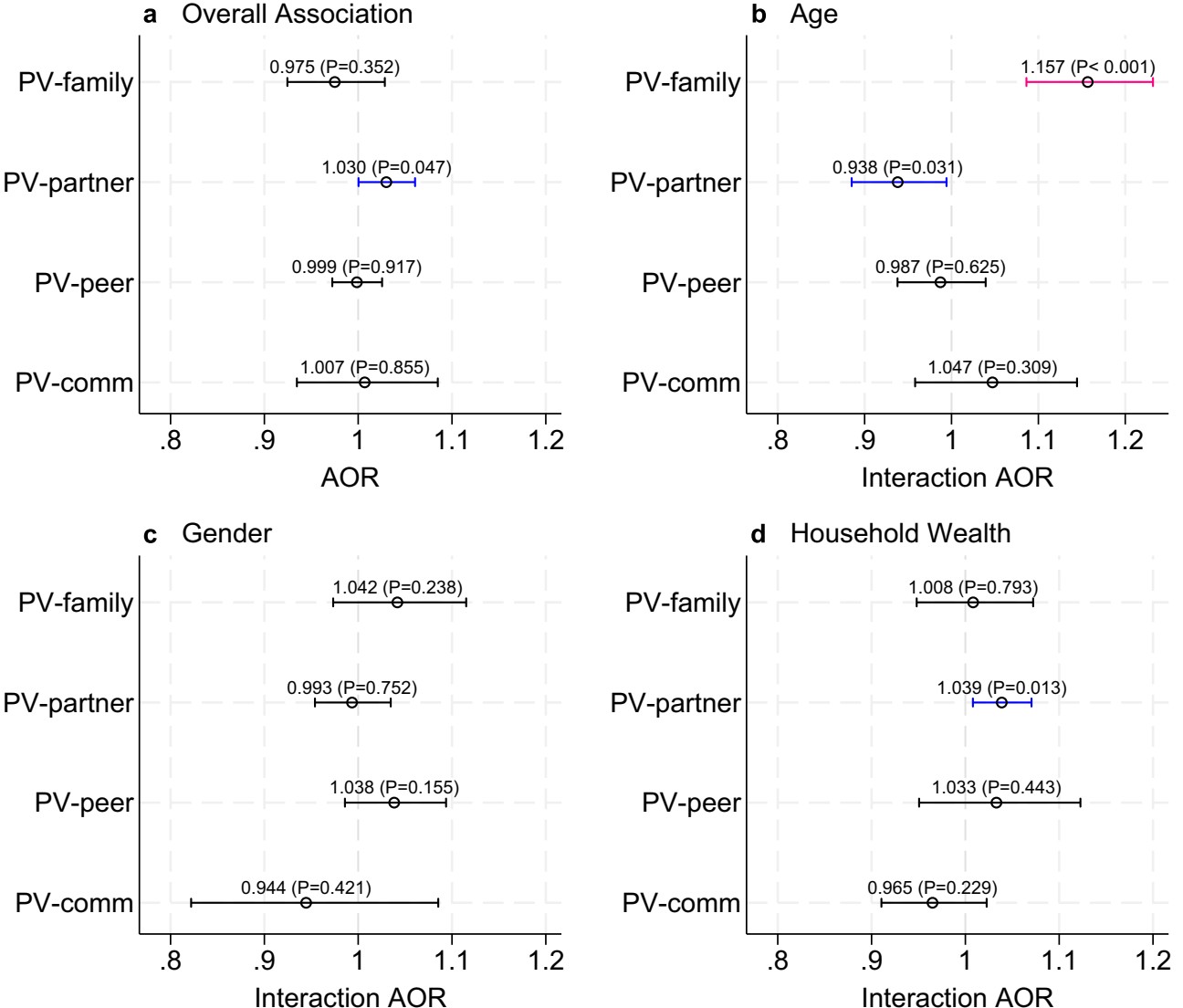

**Fig. 5 | Association between exposure to political violence and physical violence against children and youth by perpetrator type. a** Adjusted odds ratios (AOR) for the overall association between lifetime exposure to political violence and past-year physical violence (PV) from PV-family: family members; PV-partner: PV from partner; PV-peer: PV from peer; PV-comm: PV from adult in the community. The sample size is 35,439, while for PV-partner it is 21,495. **b–d** Interaction effects (Interaction AOR) examining heterogeneity in these associations by youth (18–24 years) vs. children (13–17 years) (**b**); female vs. male respondents (**c**); and lowest-low-middle vs. high-highest wealth quintiles (**d**). An AOR > 1 indicates that the association between PolV and VAC is stronger for the specified subgroup (youth, females, poorest). Blue and red bars indicate odds ratios that are statistically significant at $p < 0.05$ and $p < 0.01$, respectively. Robust standard errors clustered at the administrative division level. The bars indicate 95% confidence intervals of the estimated coefficient of interest (indicated by a small circle – the actual coefficient and its corresponding *P*-value are also listed). Full interaction results are available in Supplementary Table 6a–d.

adolescents, and young adults, long-term exposure is significantly associated with subsequent increased risk of VAC. Specifically, political violence exposure over a 15-year period increases the likelihood of experiencing subsequent emotional violence at home, physical violence from an intimate partner, and multiple instances of interpersonal violence. We also find that children, adolescents, and young adults from poorer households are at an increased risk of interpersonal violence in political violence contexts.

Our finding that emotional violence is raised after long-term exposure to political violence is robust across all types of political violence (battles, explosions, violence against civilians, riots and strategic developments) and it is the only outcome that shows such robustness. This is interesting in that emotional violence is less well researched as a form of violence against children, adolescents and young adults. At the same time, recent research highlights that

emotional violence is associated with a range of adverse health outcomes like depression, migraine and drug use[26].

The 15-year political violence index captures the lasting consequences at the household and community levels of past political violence such as parental trauma, depleted assets, weakened policing and changed social norms. These legacies continue to shape the lives of adolescents and young adults even if, for some younger respondents, the original events occurred before they were born[27]. Evidence from multiple conflict settings shows that the burdens of political violence are transmitted across generations through intertwined biological, psychological, and economic channels[28]. Our analysis highlights important correlations between past PolV and current VAC. Our findings are a call to study these associations in more detail.

Building on the early findings in this emerging literature[29], our findings are broadly consistent with localised case studies from

settings with political violence. For example, exposure to political violence in the occupied Palestinian territories is associated with increased odds of intimate partner violence victimisation in women[30]. Our results are also consistent with evidence from the USA, which shows that neighbourhood-level exposure to violent crime increases the risk of interpersonal violence within families, including intimate partner violence[31].

While we are not aware of any rigorous studies formally testing competing mechanisms by which exposure to political violence affects VAC, our results point to several possibilities, which are not mutually exclusive. The postulated mechanisms are (i) culture of insecurity, (ii) social learning of violence, (iii) social norms, (iv) socio-economic hardship, and (v) the consequence of the breakdown in civil society and government service provision and support. We will review these in turn.

First, political violence may create a culture of insecurity where people feel their physical safety is at risk, which may in turn increase stress, depression, and anxiety[21,32,33]. These effects may be exacerbated in patriarchal contexts where hegemonic masculinities position male caregivers and partners as protectors and providers, with political violence interfering with the ability to perform these roles[34]. Our heterogeneity analysis, showing that household wealth significantly mediates the relationship between political violence and sexual violence reported by adolescents and young adults, is consistent with this possibility. Future research should further explore this pathway.

Second, exposure to political violence may increase the likelihood of violence in interpersonal relationships via social learning pathways[35], where future perpetrators observe violence as a mechanism to resolve conflict or achieve desired actions in others[22]. We postulate that the timing of the political violence and the age of the victim both matter. On the one hand, it may be that it takes time to learn such adverse practices, as indicated by our finding that exposure to political violence has a significant long-term association with VAC. On the other hand, our finding that young adults aged 18–24 are more vulnerable to political violence relative to adolescents aged 13–17 suggests that there may be windows in the social development of young people where the consequences of outside adverse events are more strongly felt[36]. Perhaps with rising age, weakening of community and other support systems resulting from long-term political violence, have a stronger direct impact on young people, as they increasingly orient themselves away from the home. In fact, age at exposure to political violence may matter for adverse outcomes. In the case of the genocide in Rwanda, for example, age at exposure shaped attitudes to violence against children[37] and female empowerment[38].

Third, and relatedly, social norms may represent another mechanism by which political violence leads to violence against children, adolescents, and young adults, especially via gendered social norms in times of mass political violence like war. Qualitative research has identified that IPV victims postulate this mechanism in post-war settings[39,40]. During war, masculine and violent norms may normalise, with such norms then being internalised. In contrast, quantitative research on Angola has failed to identify such a link, instead finding that wartime exposure to collective gender-based violence reduces self-control among ex-combatants and thus leads to IPV[22].

Fourth, we observe that adolescents and young adults from poorer households may be more vulnerable to sexual violence. Economic hardship, weak livelihoods, or low socioeconomic status may elevate vulnerability to violence[30,41]. These findings align with evidence of cash transfers to adolescent girls reducing their vulnerability to sexual exploitation during periods of political violence[42]. In contrast, adolescents and young adults from households in lower wealth quintiles are less likely to experience emotional violence when exposed to political violence than wealthier counterparts, suggesting that wealth can both protect against and increase the risk of violence in a prolonged political violence setting. One explanation may be that families with lower socioeconomic status may be more resilient and adaptable during crises, leading to more stable parenting[43]. Wealthier families, accustomed to stability, may struggle more with sudden disruptions, leading to significant emotional and psychological distress. This is evident in the increased parental stress and its adverse effects on children in families of higher socioeconomic status during the COVID-19 pandemic as well as the higher incidence of depressive symptoms among wealthier households in Nigeria when exposed to political violence[44,45]. However, further research is needed to understand these conflicting mechanisms.

Fifth, in times of crisis and political violence (and especially if this lasts a long time), fiscal resources may erode as tax compliance decreases, and spending may be reprioritised towards a security economy and away from children, adolescents and young adults.

Our analysis has several strengths. We use gold-standard data on VAC from surveys, where adolescents and young adults self-report their experiences, compared to administrative data, which underestimates prevalence[46]. However, more stigmatised forms of violence, such as sexual violence, may still be under-reported. Similarly, and in contrast to the Uppsala conflict event database, which focuses on armed conflicts and their fatalities[47], the comprehensive ACLED project also covers lower-intensity political violence events. Our findings remain robust across alternative model specifications, confirming that the association between political violence and VAC is driven by the violence of events rather than by general political instability.

Our analysis has some limitations. First, given the use of observational data, we cannot establish causality between political violence and violence against adolescents and young adults. Having said that, the strict temporal sequence of the two phenomena is suggestive. Second, given the way questions about violence are asked in the VACS, we are unable to adjust for experiences of interpersonal violence prior to the last 12 months. Third, the VACS data available for public use do not include the exact GPS coordinates, to protect the identity of respondents self-reporting their experience of violence. This meant we had to aggregate ACLED data to larger area units to match the two datasets. We cannot, therefore, control for the exact distance between the exposed person and the location of the violent event, which is a common approach in the conflict literature[48]. Our estimates are likely to measure the lower bound of the indirect associations with political violence – that is, for the case of living in a relatively large area with past political violence, sometime in a long period of time, rather than the immediate, direct exposure to a nearby event. Additionally, because we lack the respondents' location histories for the 15-year window, some residents in currently "peaceful" locations may have lived in conflict areas in the past and vice versa, which could bias our estimates. However, a review of displacement datasets and VACS sampling notes shows that seven of the nine study countries experienced little or no conflict-related migration during our exposure windows, and the two with recent displacement (Nigeria and Mozambique) excluded the most insecure clusters. Fourth, our methodology precludes identifying specific temporal windows of child or adolescent development in which political violence exerts particularly adverse impacts. We do learn, however, that the long-term historical context of political violence matters for an adolescent's or young person's current experience of VAC.

The public health implications of this study are compelling. Our results suggest the need for additional attention to children, adolescents and young adults living in areas shaped, even many years prior, by political violence. There are examples of recent parenting programmes to reduce domestic violence that could be adapted and considered for implementation in contexts of political violence[49]. There is a need to develop further school-based programming for conflict-affected areas, and there is little evidence of the effectiveness of any school-based violence prevention programmes in any area with past political violence[50]. Programmes to prevent intimate partner

violence may hold special promise, as these programmes can often simultaneously reduce IPV and VAC within the same households[51], and examples exist of interventions which can be deployed in conflict and post-conflict contexts[29,52].

## Methods

### Data sources

As a secondary data analysis, our study did not require approval by an ethics committee.

For our analysis, we combine secondary data from two main sources using Stata 18: the Violence Against Children and Youth Surveys (VACS)[53] and the Armed Conflict Location and Event Data (ACLED)[54,55] (see Supplementary Methods). We believe such matching has not been done previously.

**VACS.** The VACS are nationally representative household surveys focusing on adolescents and young people aged 13-24 years[56]. The VACS are conducted and collected by national governments with technical assistance provided by the US Center for Disease Control and Prevention (CDC) and with global coordination provided by the Together for Girls (TfG) partnership, in turn hosted by the United Nations. Each survey adopts a three-stage cluster split-sample design that selects one respondent per household and assigns male and female interviews to different enumeration areas, ensuring boys and girls are never interviewed in the same geographical areas. The survey systematically records self-reported experiences of physical, sexual, and emotional violence during the past 12 months preceding the survey, along with lifetime exposure, which also captures experiences before respondents turn 18 years old. The gender of respondents is self-reported; VACS questionnaires only contain categories for male and female respondents. VACS data are available for research on request, free of charge. We used all countries VACS that had been made available to us by mid-2024 and which were also represented in the ACLED, resulting in a sample of nine African countries. The VACS geo-reference codes for each country cover administrative areas or units that vary in definition and population size (e.g., regions in Malawi; provinces in Cote d'Ivoire, local government areas in Nigeria, counties in Kenya, or districts in Zambia). Across the nine countries we study, there are 723 sub-national administrative units which range in size from 8.60 to 84,750 square kilometres (mean=11,073; s.d.=17,744), with populations ranging from 14,460 to 6,321,017 (mean=694,333; s.d.=1,162,791). The mean number of VACS interviews per administrative unit is 49.01 (s.d.=114.9).

Although the VACS aim to be nationally representative for both sexes, this is not the case for all VACS countries. We find from each country's technical report that country teams set different sample targets for boys and girls with some countries deliberately choosing to oversample girls, who may be perceived to face higher risks of violence, and some countries experiencing lower male survey completion rates. As a result, in four of the nine VACS, surveys ended up skewed toward girls: roughly 90% female in Zimbabwe, 80% in Namibia, 70% in Mozambique and 63% in Kenya, bringing the overall female share to 67%. In our models, we apply the official VACS weights, which adjust for the split-sample probabilities and differential non-response and include respondent sex as a covariate[56].

**ACLED.** ACLED codes local political violence and protest events worldwide, assembled from more than 5000 publicly available news sources, with temporal and precise geographic identifiers[55]. ACLED records the date and exact GIS location of political violence events. ACLED has been coding and aggregating political violence and protest events for many years, with coverage for African countries extending back to 1997. The data collected by ACLED are freely available for research.

**Merging ACLED and VACS.** All VACS from countries in the ACLED database were included and matched with ACLED data from the 15-year period preceding each VACS 12-month recall period. We did so by mapping the ACLED data into the VACS data by matching into the smallest administrative unit common to both datasets (see Supplementary Table 1). The merged dataset comprises VACS collected between 2013 and 2019, covering 35,439 individuals aged 13-24 from nine African countries (Côte d'Ivoire, Kenya, Malawi, Mozambique, Namibia, Nigeria, Uganda, Zambia, and Zimbabwe). Together, these VACS data represent about 30% of Africa's population.

### Measures

**Violence against children, adolescents and young adults.** The VACS measure several dimensions of violence (see Supplementary Table 2). VACS defines physical violence (PV) as acts that include slapping, pushing, punching, kicking, choking, or using a weapon to threaten or harm. The VACS questionnaires also ask about PV perpetrators, where options include family members, peers, adults in the community, and, where relevant, intimate partners. Sexual violence (SV) is defined as any unsolicited sexual act, such as touching, attempted forced sex, coerced sex, or physically forced sex, regardless of perpetrator group. Emotional violence (EV) covers verbal abuse and statements such as 'you are not loved' or the use of ridicule by family members, an intimate partner, or a peer. However, to maintain comparability across surveys, perpetrators of EV are restricted to family members.

Our main analysis focuses on the experience of VAC during the last 12 months, with separate analysis for PV, SV and EV. We also generate a variable for experience of at least one type of violence as "any violence" (AV) in the last 12 months. Our "multiple violence" (MV) variable captures a respondent's experience of more than one type of VAC in the last 12 months. It is worth noting that in Malawi, Nigeria and Zambia, if the question of having ever experienced the relevant type of VAC was answered as 'No', the question about having experienced that type of violence in the last 12 months was skipped. In these cases, we coded the response to the last 12-month question as 'No' (see Supplementary Methods).

**Political violence.** In ACLED, political violence events are categorized as battles (violent clashes between armed groups), explosions/remote violence (bombings or missile attacks), violence against civilians (targeted attacks on unarmed people) and riots (spontaneous, violent demonstrations)[55]. We also consider strategic developments, which capture contextually important developments. ACLED also codes protests (a non-violent form of political unrest), which we use in our robustness checks. We measure political violence by aggregating the number of political violence events in each sub-national administrative area within each of the nine VACS countries, adding all PolV for the past one, three, five, and fifteen years prior to the last year reference period of the relevant VACS. This ensures that our PolV indicators have a reference period that always precedes children's reported VAC experience. We first consider an aggregate index for all five categories of political violence events and then consider each category separately. We weight the political violence indicators by the population size of each administrative area in the year closest to the VACS survey to make indicators comparable across space within and between countries. We also consider an alternative indicator of political violence, fatalities, in our robustness check. Fatalities are measured as the cumulative number of conflict-related deaths recorded in each sub-national administrative unit during the 15 years preceding the survey's 12-month recall period, expressed per 100,000 population.

**Covariates.** Data on covariates are from the VACS and include age of the respondent (that is the adolescent or the young adult), gender,

school attendance, marital status and recent work history. We also used information on household assets to construct a standardised index of household wealth[57].

## Statistical analyses
We estimate Eq. (1)

$$logit\left(Pr\left(VAC_{ijk,v}=1\right)\right) = \alpha_v + \beta_{vst}PolV_{j,st} + \beta_{vlt}PolV_{j,lt} + \gamma_{vn}X_{ijkn} + \delta_k$$

(1)

with the VAC variable on the left-hand side and the PolV indicator on the right-hand side. The sub-index $i$ refers to the individual child, $j$ refers to the location, such as the district (administrative area) of residence, while $k$ represents the country. $v$ captures the type of VAC being examined: PV, SV, EV, AV or MV. For PolV, we use each of the five categories of violence events or their aggregate. The index $t$ differentiates short-term ($st$) and long-term past political violence ($lt$). $n$ refers to the vector of covariates $X$, including age, gender, marital status, school attendance, recent work history and relative household wealth: these are considered exogenous to political violence at the respondent level. $\delta_k$ are country-specific fixed effects. We estimate logistic regressions with country-specific fixed effects to account for unobserved heterogeneity. Standard errors are clustered at the administrative area level to adjust for within-cluster correlation. Population sample weights ensure that estimates are nationally representative, which also corrects for the over-sampling of female respondents in Kenya, Mozambique, Namibia, and Zimbabwe.

**Heterogeneity.** The association between political violence and the risk of interpersonal violence among adolescents and young adults are likely to vary across different subgroups. To account for possible heterogeneity, we interact PolV with dummy variables for age (13 to 17 years versus 18 to 24 years), gender (male versus female) and household wealth (three lower versus two upper quintiles) to estimate Eq. (2):

$$logit\left(Pr\left(VAC_{ijkv}=1\right)\right) = \alpha_v + \beta_{vlt}PolV_{j,lt} + \gamma_{vn}X_{ijkn} + \psi_{vp}(PolV_j X D_{p,i}) + \delta_k$$

(2)

where $D_{p,i}$ is age, gender, and wealth with $p$ indexing these dummy variables. The coefficients $\psi_{vp}$ capture how the role of PolV changes by subgroup.

We also study PV by perpetrator, noting that PV is the only type of VAC that can be disaggregated by perpetrator across all nine countries.

**Robustness.** To check for robustness, we undertake five additional tests: (i) we estimate the model replacing political violence events with peaceful protests as a placebo; (ii) we estimate the model without any or only with some covariates; (iii) we estimate the model without household wealth to exclude the possible adverse impacts of PolV on wealth; (iv) we use an alternative measure of PolV, namely political violence fatalities; and (v) we split the 15-year PolV exposure period into two non-overlapping periods, namely one to five years (which we had estimated in Eq. (1) already) and six to fifteen years.

## Reporting summary
Further information on research design is available in the Nature Portfolio Reporting Summary linked to this article.

## Data availability
For this secondary data analysis project, the merged scientific use file based on VACS[53] and ACLED[54,55] including de-identified participant data underlying the analysis are available on the ReShare web site of the UK Data Service at https://doi.org/10.5255/UKDA-SN-858336. The raw data from VACS and ACLED are not included due to third-party licence restrictions. To obtain raw VACS data, contact the global partnership Together for Girls at https://www.togetherforgirls.org/en/about-the-vacs and the US Centers for Disease Control and Prevention (CDC) at https://www.cdc.gov/violenceprevention/childabuseandneglect/vacs/index.html, who provide technical advice on VACS to Together for Girls. To obtain raw data from ACLED, contact https://acleddata.com/.

## Code availability
The final statistical code to replicate the analysis is available on the ReShare web site of the UK Data Service at https://doi.org/10.5255/UKDA-SN-858336.

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

## Acknowledgements

This research was funded by the Economic and Social Research Council (ESRC), UK – project number ES/X00192X/1 (MV, OF, KD, VI & TB) and by the Deutsche Forschungsgemeinschaft (DFG, German Research Foundation) – project number BR 3614/4-1 (TB). The funders had no role in study design, data collection, data analysis, data interpretation, writing or the decision to submit the manuscript.

## Author contributions

M.V.: Conceptualization; methodology; data access and verification; formal analysis; investigation; writing – original draft, review & editing; supervision; funding acquisition. O.F.: Data access, verification & curation; formal analysis; investigation; visualisation; writing – original draft, review & editing. K.D.: Investigation; writing – review & editing; funding acquisition. V.I.: Investigation; writing – original draft, review & editing; supervision. T.B.: Conceptualization; methodology; investigation; writing – original draft, review & editing; supervision; funding acquisition.

## Funding

## Competing interests

The authors declare no competing interests.
