## [Transparent Peer Review file · Nature Communications]

Past political violence and interpersonal violence against children and youth in Africa

Corresponding Author: Professor Tilman Brück

Version 0:

Reviewer comments:

Reviewer #1

(Remarks to the Author)

The manuscript presents a robust analysis of the impact of exposure to political violence on various forms of violence against children, using a novel, linked dataset. The analytical approach is sound and is further strengthened by employing several sensitivity analyses for robustness. The results offer a significant contribution to the existing evidence on the links between political violence and interpersonal violence for children, most importantly, by looking at the impacts for multiple forms of violence, perpetrators, and levels of wealth. We offer suggestions below, which believe may strengthen this manuscript.

1. The introduction clarifies the potential pathways through which exposure to armed conflict may impact subsequent violence against children, but no summary of the existing empirical evidence on these relationships is presented. While not solely focused on children, there is a relatively substantive body of literature that looks at the increased risk of intimate partner violence for those affected by conflict (and some of this literature focuses on the 13-24-year-old age group). There are also a few studies that examine the impact of conflict exposure on other forms of violence against children (see Malcolm, Diwakar, and Naufal, 2020 in the Journal of Conflict Resolution and Stark et al. 2023 in Child abuse and neglect, for example). The introduction could benefit from a more comprehensive summary of the existing relevant literature and how this study adds to this evidence base.
2. The current Methods section jumps back and forth between descriptions of the datasets, dataset linking approach, and measures. The section needs to be organized into sub-sections to ease readability and comprehension. A suggested restructuring: Data sources: A summary of the ACLED and VACS datasets, separately (including sampling approaches, data collection procedures), as well as a description of how the datasets were linked; Measures: A description of the measures employed from each dataset; Analysis.
3. Page 6: Why does the final sample comprise 67% female respondents? Aren't the VACS generally equal parts male and female respondents?
4. Discussion: The authors note that "young adults aged 18-24 are more vulnerable to the effects of political violence relative to children aged 13-17," but do not discuss this point further. This is an interesting finding, especially given that the outcome of interest is violence in the past 12 months rather than lifetime violence AND given that increases in physical violence by family members—and not by intimate partners—are higher for this older group. What are the hypothesized mechanisms for this difference?
5. The finding that emotional violence increased for all political violence categories is important, and worth expanding on in the Discussion. Emotional Violence is often overlooked but is linked to many mental and physical health outcomes. It would be helpful to elucidate the consequences and lifetime implications of this type of violence.
6. Also alluded to in the Discussion, but not mentioned explicitly are the role of social norms. In addition to social learning platforms, there is some evidence that men who feel emasculated and disempowered due to political or communal violence may reassert power in the private sphere, exacerbating IPV and child maltreatment. There is a good deal of literature on violence, power, and shifting social norms that could be included in the Discussion.

7. Finally, the authors note as a limitation that they cannot control for the exact distance from the political event. An additional limitation should be added around the inability to know whether the respondent lived in the same location during the timing of the political violence, especially given that the authors measure political violence as far back as 15 years. Given the correlation between political violence and forced displacement, it is likely that some of the respondents living in non-violent geographical areas during the VAC interview were exposed to political violence in the last 15 years and vice versa. This limitation may result in an underestimate of effect sizes.

-

(Remarks on code availability)

Reviewer #2

(Remarks to the Author)

SUMMARY

This paper examines the association between political violence and violence against children in nine countries in Sub-Saharan Africa. The authors combine nationally representative surveys on violence against children (VACS) with the incidence of political violence in the region of residence of survey respondents from the ACLED database. They find that political violence is associated with violence against children in the long term (defined as exposure in the past 15 years), but not in the short term (defined as exposure in the past 5 years). Additionally, certain groups are more vulnerable to the increasing effects of political violence.

OVERALL EVALUATION

The paper addresses an important and relevant research question and enhances our understanding of factors that may contribute to violence against adolescents and younger adults. The empirical analysis utilizes the appropriate data sources and is competently executed.

MAIN COMMENTS

1. The authors write that they study violence against children and youth, but their sample includes individuals aged 13-24, and interpersonal violence is defined as violence that occurred in the past 12 months before the interview. It seems that the focus is on adolescents and youth rather than children and youth.
2. Long-term exposure is defined as exposure in the past 15 years, which includes episodes of political violence that happened when the youngest adolescents in the sample (ages 13 and 14) were not born yet. Maybe the authors are thinking that the adolescents' parents are affected and then the adolescents are affected through an intergenerational transmission, but if that is the case, it should be clarified in the Discussion section.
3. If I understood correctly, because the data is repeated cross-section, the empirical analysis conflates the time since the political violence occurred (for instance, past 15 years for the long-term exposure and past 5 years for the short-term exposure) with the individuals' age at exposure (for instance, current age of the respondents at the time of the interview minus 15 years for the long-term exposure). Then maybe one potential interpretation of the results is that exposure before birth, during early childhood and childhood (age -2 to age 9, obtained by subtracting 15 years from age 13 to 24) matters more than exposure during adolescence (age 8 to 19, obtained by subtracting 5 years from age 13 to age 24) for increasing the odds of being subjected to interpersonal violence.
4. Political violence may cause migration, which may potentially lead to endogenous sample selection in the sample and selection bias in the estimates. If children with a higher risk of interpersonal violence are more likely to stay in the provinces with high political violence, and children with lower risk are more likely to move to the low-violence regions, this may lead to upward bias in the estimates. Do the VACS surveys provide any information on migration?
5. The measure of long-term exposure (past 15 years) includes the measure of short-term exposure (past 5 years). Is there a way to differentiate between the two? For example, the authors could include in the same regression exposure in the past 5 years (t-1 to t-5) and exposure in the 10 preceding years (t-6 to t-15).
6. I don't think the authors provide a compelling explanation for why only the long-term exposure matters and the short-term exposure does not. Could the authors elaborate more on possible explanations, without indulging in speculation?

(Remarks on code availability)

Reviewer #3

(Remarks to the Author)
REVIEW OF PAPER

The title of the reviewed paper is 'Past political violence and interpersonal violence against children and youth: Micro-level evidence from nine African countries'.

Summary

This paper presents the findings from a groundbreaking study that examines the relationship between past political violence and subsequent interpersonal violence against children and youth. The authors investigate how exposure to political violence may increase the risk of various forms of interpersonal violence, focusing specifically on children and youth. This analysis utilizes nationally representative data from Côte d'Ivoire, Kenya, Malawi, Mozambique, Namibia, Nigeria, Uganda, Zambia, and Zimbabwe. The study combines multi-country, cross-sectional microdata from the Violence against Children and Youth Surveys (VACS) with conflict event data from the Armed Conflict Location and Event Data (ACLED), using geographic and temporal identifiers for correlation. Logistic regression analyses are employed to assess the link between prior exposure to political violence and the increased likelihood of children and youth experiencing physical, emotional, or sexual violence. First, the study explores whether short-term or long-term exposure to political violence heightens the chances of children and youth encountering recent incidents of physical, sexual, or emotional abuse. The authors also investigate whether the adverse effects of exposure to political violence differ across various categories of such violence. These findings underscore the necessity of considering the political and historical context when devising strategies to prevent and address violence against children and youth, whether in families, schools, or early intimate relationships. The results pave the way for future research and initiatives aimed at achieving Sustainable Development Goal 16.2, which focuses on the prevention of violence against children and youth.

Methodology

The study used a quantitative, cross-sectional, and observational design to analyze secondary data. The Violence Against Children Surveys (VACS) were conducted at a single point in time, which differentiates them from longitudinal studies. Researchers observed and analyzed existing data, focusing on variables like experiences of violence and exposure to conflict, using datasets from VACS and the Armed Conflict Location & Event Data Project (ACLED) rather than collecting primary data. All Violent Against Children and Youth Surveys (VACS) from countries in the Armed Conflict Location and Event Data (ACLED) database were included and matched with ACLED data from the 15 years preceding each VACS. The resulting merged dataset includes VACS collected between 2013 and 2019, covering 35,437 individuals aged 13 to 24 from nine countries: Côte d'Ivoire, Kenya, Malawi, Mozambique, Namibia, Nigeria, Uganda, Zambia, and Zimbabwe. Together, these VACS data represent approximately 30% of Africa's population. The Violent Against Children and Youth Surveys (VACS) are nationally representative household surveys that focus on children and youth aged 13 to 24 years. The Armed Conflict Location and Event Data (ACLED) codes conflict events across countries, using information gathered from publicly available news sources, along with temporal and precise geographic identifiers.

Strengths of the Paper

- In citing other works as evidence or presenting factual information, this paper acknowledges the authors and studies referenced.
- Methodology and Discussion were thoroughly explained.
- Sufficient Use of Visuals (Tables, Graphs, and Charts) made the data easy to interpret with visual representation.
- The authors identified and stated several important topics that need further research related to the subject matter.
- The results were rigorously tested and subjected to a placebo test to verify their accuracy and reliability.
- The secondary data (VACS and ACLED) used is valid and reliable.

Shortcomings of the Paper

- In the Introduction, the terms in the topic are not properly expanded with case studies and scenarios.
- The authors do not explain the chosen methodology or its limitations.
- The paper does not provide any recommendations for resolving these issues.
- The author does not outline the limitations and misconceptions in research concerning the subject.
- The data is cross-sectional and, as such, cannot establish causality; it merely highlights correlations. While exposure to conflict is associated with an increased risk of violence, one cannot conclude that one directly causes the other.
- The paper does not explore the various fields that relate to or influence the topic.
- A layperson may find it difficult to understand the paper, particularly the results and discussion sections.

Recommendations

- The limitations in causality, potential biases, and lack of qualitative depth must be acknowledged when interpreting results.
- Detail how the findings can assist policymakers, practitioners, or local communities in advancing Sustainable Development Goal 16.2.

(Remarks on code availability)
REVIEW OF PAPER

The title of the reviewed paper is 'Past political violence and interpersonal violence against children and youth: Micro-level evidence from nine African countries'.

Summary

This paper presents the findings from a groundbreaking study that examines the relationship between past political violence and subsequent interpersonal violence against children and youth. The authors investigate how exposure to political violence may increase the risk of various forms of interpersonal violence, focusing specifically on children and youth. This analysis utilizes nationally representative data from Côte d'Ivoire, Kenya, Malawi, Mozambique, Namibia, Nigeria, Uganda, Zambia, and Zimbabwe. The study combines multi-country, cross-sectional microdata from the Violence against Children and Youth Surveys (VACS) with conflict event data from the Armed Conflict Location and Event Data (ACLED), using geographic and temporal identifiers for correlation. Logistic regression analyses are employed to assess the link between prior exposure to political violence and the increased likelihood of children and youth experiencing physical, emotional, or sexual violence. First, the study explores whether short-term or long-term exposure to political violence heightens the chances of children and youth encountering recent incidents of physical, sexual, or emotional abuse. The authors also investigate whether the adverse effects of exposure to political violence differ across various categories of such violence. These findings underscore the necessity of considering the political and historical context when devising strategies to prevent and address violence against children and youth, whether in families, schools, or early intimate relationships. The results pave the way for future research and initiatives aimed at achieving Sustainable Development Goal 16.2, which focuses on the prevention of violence against children and youth.

Methodology

The study used a quantitative, cross-sectional, and observational design to analyze secondary data. The Violence Against Children Surveys (VACS) were conducted at a single point in time, which differentiates them from longitudinal studies. Researchers observed and analyzed existing data, focusing on variables like experiences of violence and exposure to conflict, using datasets from VACS and the Armed Conflict Location & Event Data Project (ACLED) rather than collecting primary data. All Violent Against Children and Youth Surveys (VACS) from countries in the Armed Conflict Location and Event Data (ACLED) database were included and matched with ACLED data from the 15 years preceding each VACS. The resulting merged dataset includes VACS collected between 2013 and 2019, covering 35,437 individuals aged 13 to 24 from nine countries: Côte d'Ivoire, Kenya, Malawi, Mozambique, Namibia, Nigeria, Uganda, Zambia, and Zimbabwe. Together, these VACS data represent approximately 30% of Africa's population. The Violent Against Children and Youth Surveys (VACS) are nationally representative household surveys that focus on children and youth aged 13 to 24 years. The Armed Conflict Location and Event Data (ACLED) codes conflict events across countries, using information gathered from publicly available news sources, along with temporal and precise geographic identifiers.

Strengths of the Paper

- In citing other works as evidence or presenting factual information, this paper acknowledges the authors and studies referenced.
- Methodology and Discussion were thoroughly explained.
- Sufficient Use of Visuals (Tables, Graphs, and Charts) made the data easy to interpret with visual representation.
- The authors identified and stated several important topics that need further research related to the subject matter.
- The results were rigorously tested and subjected to a placebo test to verify their accuracy and reliability.
- The secondary data (VACS and ACLED) used is valid and reliable.

Shortcomings of the Paper

- In the Introduction, the terms in the topic are not properly expanded with case studies and scenarios.
- The authors do not explain the chosen methodology or its limitations.
- The paper does not provide any recommendations for resolving these issues.
- The author does not outline the limitations and misconceptions in research concerning the subject.
- The data is cross-sectional and, as such, cannot establish causality; it merely highlights correlations. While exposure to conflict is associated with an increased risk of violence, one cannot conclude that one directly causes the other.
- The paper does not explore the various fields that relate to or influence the topic.
- A layperson may find it difficult to understand the paper, particularly the results and discussion sections.

Recommendations

- The limitations in causality, potential biases, and lack of qualitative depth must be acknowledged when interpreting results.
- Detail how the findings can assist policymakers, practitioners, or local communities in advancing Sustainable Development Goal 16.2.

Reviewer #4

(Remarks to the Author)

(Remarks on code availability)

Version 1:

Reviewer comments:

Reviewer #1

(Remarks to the Author)

Thanks to the authors for their careful attention to our feedback. I have reviewed their revisions and am satisfied that they have attended to our requested revisions.

(Remarks on code availability)

Reviewer #2

(Remarks to the Author)

All my comments were addressed in the revision. I enjoyed reading the revised draft.

(Remarks on code availability)

Past political violence and interpersonal violence against children and youth: Micro-level evidence from Africa

Reply to Reviewers

7 August 2025

Comments are in *italics*.

Replies are in blue.

Reviewer #1

The manuscript presents a robust analysis of the impact of exposure to political violence on various forms of violence against children, using a novel, linked dataset. The analytical approach is sound and is further strengthened by employing several sensitivity analyses for robustness. The results offer a significant contribution to the existing evidence on the links between political violence and interpersonal violence for children, most importantly, by looking at the impacts for multiple forms of violence, perpetrators, and levels of wealth. We offer suggestions below, which (we) believe may strengthen this manuscript.

Thank you for your assessment of our paper, which we greatly appreciate. We would like to clarify that we estimate *associations* between past exposure to political violence and recent violence against adolescents and young adults, not *impacts* of past exposure to political violence on violence against adolescents and young adults.

1. The introduction clarifies the potential pathways through which exposure to armed conflict may impact subsequent violence against children, but no summary of the existing empirical evidence on these relationships is presented. While not solely focused on children, there is a relatively substantive body of literature that looks at the increased risk of intimate partner violence for those affected by conflict (and some of this literature focuses on the 13-24-year-old age group). There are also a few studies that examine the impact of conflict exposure on other forms of violence against children (see Malcolm, Diwakar, and Naufal, 2020 in the Journal of Conflict Resolution and Stark et al. 2023 in Child abuse and neglect, for example). The introduction could benefit from a more comprehensive summary of the existing relevant literature and how this study adds to this evidence base.

Thank you for these comments and valuable suggestions about literature to cite. We have expanded the introduction section (on pages 3-5) and added these suggestions and other relevant references to expand our review of relevant literature. We agree that the literature examining the relationship between political violence and violence against children and youth is limited and agree that there are adjacent literatures that we can further summarise and which we have now done. We have now organised this section

according to various mechanisms/pathways by which political violence and violence in childhood may be associated and draw on papers looking at violent conflict and intimate partner violence against women, parenting, children's mental health in conflict settings, as well as the (lack of) functioning of child protection and other systems in violent contexts.

2. The current Methods section jumps back and forth between descriptions of the datasets, dataset linking approach, and measures. The section needs to be organized into sub-sections to ease readability and comprehension. A suggested restructuring: Data sources: A summary of the ACLED and VACS datasets, separately (including sampling approaches, data collection procedures), as well as a description of how the datasets were linked; Measures: A description of the measures employed from each dataset; Analysis.

Thank you for this very helpful reorganization suggestion. We have now restructured the Methods section following your suggestion and using the proposed use sub-headings: "Data sources" subsection now describes the two datasets used, the VACS and the ACLED, including sampling approaches and data collection procedures while also explaining the data-merging procedure; "Measures" subsection discusses the definitions of the variables used and constructed; and "Statistical analyses" subsection covers the main models, heterogeneity analysis, and robustness checks. Please find the changes made to the Methods section on pages 12 to 16 in the manuscript.

3. Page 6: Why does the final sample comprise 67% female respondents? Aren't the VACS generally equal parts male and female respondents?

Although the VACS aim to be nationally representative for both sexes, this is not the case for all VACS countries. Each country's technical reports show that country teams set different sample targets for boys and girls with some countries deliberately choosing to oversample girls, who may be perceived to face higher risks of violence. As a result, samples for four of the nine surveys ended up enumerating a higher proportion of girls: roughly 90% female in Zimbabwe, 80% in Namibia, 70% in Mozambique and 63% in Kenya, bringing the overall female share to 67%. In our models, we apply the official VACS weights, which adjust for the split-sample probabilities along with differential non-response for boys and girls: we also include respondent sex as a covariate (see Nguyen et al. 2019 (56)). See the revised Methods section for more details.

4. Discussion: The authors note that "young adults aged 18-24 are more vulnerable to the effects of political violence relative to children aged 13-17," but do not discuss this point further. This is an interesting finding, especially given that the outcome of interest is violence in the past 12 months rather than lifetime violence AND given that increases in physical violence by family members—and not by intimate partners—are higher for this older group. What are the hypothesized mechanisms for this difference?

Thank you for emphasizing this important observation and for suggesting more discussion of the finding that "young adults aged 18-24 are more vulnerable to the effects

of political violence relative to children aged 13-17", which indeed, and as you rightly point out, should have been given more attention in the original manuscript. We note that we study reported VAC in the last twelve months and if this may be associated with exposure to political violence over the last 15 years. We have now added a discussion of this finding on pages 9-10 in the context both of the "social learning of violence" pathway and through the weakening of community and other support systems resulting from long-term political violence.

5. The finding that emotional violence increased for all political violence categories is important, and worth expanding on in the Discussion. Emotional Violence is often overlooked but is linked to many mental and physical health outcomes. It would be helpful to elucidate the consequences and lifetime implications of this type of violence.

Thank you for this excellent suggestion. We added a short discussion of this finding on page xxx, which is very robust across all types of political violence. We also now discuss our finding in light of a paper in *Nature Human Behaviour*, which highlights the point you raised, namely that there are many costs to experiencing emotional violence (26).

6. Also alluded to in the Discussion, but not mentioned explicitly are the role of social norms. In addition to social learning platforms, there is some evidence that men who feel emasculated and disempowered due to political or communal violence may reassert power in the private sphere, exacerbating IPV and child maltreatment. There is a good deal of literature on violence, power, and shifting social norms that could be included in the Discussion.

This is an important topic and indeed there is a debate in the literature around gender, masculinity, norms and war, especially regarding IPV. However, few papers test such a relation with IPV explicitly, which is probably also related to data limitations. With regards to VAC, we are not aware of any papers reviewing or even formally testing this association. We acknowledge and agree with the merit of this suggestion and have expanded the discussion on pages 9-10 accordingly.

7. Finally, the authors note as a limitation that they cannot control for the exact distance from the political event. An additional limitation should be added around the inability to know whether the respondent lived in the same location during the timing of the political violence, especially given that the authors measure political violence as far back as 15 years. Given the correlation between political violence and forced displacement, it is likely that some of the respondents living in non-violent geographical areas during the VAC interview were exposed to political violence in the last 15 years and vice versa. This limitation may result in an underestimate of effect sizes.

Thank you for drawing our attention to this limitation. We have now added to the manuscript that, because we lack the respondents' location histories for the 15-year window, some residents in currently "peaceful" locations may have lived in areas affected by political violence in the past and vice-versa (see page 12).

Such a misclassification could bias our estimates, likely toward understating the true effect (that is our estimates are conservative). To test this formally, we would need more accurate (current and past) location data about VACS respondents. However, such data is not available for good reasons - to protect the respondents.

It is also worth emphasizing that many of the areas we study are not war zones. In fact, VACS are not typically conducted in crisis settings (at least not at the time when we accessed the VACS data). We thus do not study war but political violence, which is quite prevalent, even in countries often considered peaceful like Namibia (which is included in our sample). This concern would have been more pressing for areas experiencing more intense crises.

To more formally gauge the relevance of the issue, we reviewed IDMC/IOM displacement data (<https://www.internal-displacement.org/>) and the VACS sampling notes for every country. Namibia, Malawi, and Zambia have had no conflict-related displacement for more than two decades. Zimbabwe experienced a brief postelection population movement in mid-2008, but most households returned within months; misclassification should therefore be negligible in these four countries. In Uganda, Kenya, and Côte d'Ivoire, the last large displacement waves ended at least six years before the interpersonal-violence reference year we analyse. Specifically, the Lord's Resistance Army camps were closed by 2008, Kenya's 2007–08 IDPs had resettled by 2012, and Côte d'Ivoire's 2002–11 IDPs returned home by 2011.

Also, Nigeria (2014) and Mozambique (2019) are the only cases with considerable displacement in the three years leading up to the surveys, and both VACS intentionally skipped the most insecure clusters (three LGAs in Borno and several PSUs in Cabo Delgado).

In the VACS, only Namibia asks a migration-related question, namely “Have you ever moved to another region in your country?”. The follow-up question on the reason for moving has no answer option for conflict-induced displacement. Upon assessing the potential association between the migration indicator and VAC outcomes, we found no significant association ($p > 0.40$).

Finally, we like to point out that our approach involves matching ACLED data and VACS data by quite large administrative areas. As we say in the Methods section: “Across the nine countries we study, there are 723 sub-national administrative units which range in size from 8.60 to 84,750 square kilometres (mean=11,073; s.d.=17,744), with populations ranging from 14,460 to 6,321,017 (mean=694,333; s.d.=1,162,791). The mean number of VACS interviews per administrative unit is 49.01 (s.d.=114.9).” It is hence likely that internal displacement may have occurred within the same administrative area (such as provinces in Cote d'Ivoire or local government authorities in Nigeria). Had we been able to narrow down the location of the respondents in the VACS more accurately, then this

displacement bias might have been larger. As it is, we believe that this is not likely to shape our findings significantly.

Reviewer #2

SUMMARY

This paper examines the association between political violence and violence against children in nine countries in Sub-Saharan Africa. The authors combine nationally representative surveys on violence against children (VACS) with the incidence of political violence in the region of residence of survey respondents from the ACLED database. They find that political violence is associated with violence against children in the long term (defined as exposure in the past 15 years), but not in the short term (defined as exposure in the past 5 years). Additionally, certain groups are more vulnerable to the increasing effects of political violence.

OVERALL EVALUATION

The paper addresses an important and relevant research question and enhances our understanding of factors that may contribute to violence against adolescents and younger adults. The empirical analysis utilizes the appropriate data sources and is competently executed.

Thank you for this positive overall endorsement of our study. We would just like to clarify and add that we study multiple windows of political violence exposure (from one year till 15 years) and that we also find that some types of VAC are more highly correlated with exposure to political violence than others.

MAIN COMMENTS

1. *The authors write that they study violence against children and youth, but their sample includes individuals aged 13-24, and interpersonal violence is defined as violence that occurred in the past 12 months before the interview. It seems that the focus is on adolescents and youth rather than children and youth.*

Thank you for this comment. In our original submission, we followed the terminology consistently adopted in the VACS questionnaires and reports, which talk about “children” (13-17 years) and “youth” (18–24 years). (These ages refer to the age of respondents at the time of the interview.) We agree that this can be interpreted as somewhat misleading when considering these age brackets. In our revised paper, we have therefore adopted the terms “children, adolescents and young adults” when speaking about the group of interest generally and “adolescents” (13-17 years) and “young adults” (18–24 years) when speaking about these specific age groups. We made these changes throughout the revised manuscript.

2. Long-term exposure is defined as exposure in the past 15 years, which includes episodes of political violence that happened when the youngest adolescents in the sample (ages 13 and 14) were not born yet. Maybe the authors are thinking that the adolescents' parents are affected and then the adolescents are affected through an intergenerational transmission, but if that is the case, it should be clarified in the Discussion section.

We limit outcomes to interpersonal violence experienced in the past 12 months to ensure that the self-reported incidents occurred after the periods of exposure to political violence.

We agree that the 15-year exposure window captures events that occurred before the youngest respondents were born. Our interpretation does not assume that children are exposed in utero or that parental exposure pre-conception of their child is a key mechanism. Instead, the 15-year time horizon measures chronic exposure or the accumulated effects of conflict on parental trauma, household economic resources, local institutions and social norms.

Note also that we aggregate all political violence events during that period, not weighing events which occurred longer ago as we do not know what a reasonable decaying function (or temporal weight) would be. It is numerically unlikely that there are many political violence events prior to the birth of our youngest respondents as most of our respondents are aged 15 and older. We have added a paragraph to the Discussion on p. 9.

3. If I understood correctly, because the data is repeated cross-section, the empirical analysis conflates the time since the political violence occurred (for instance, past 15 years for the long-term exposure and past 5 years for the short-term exposure) with the individuals' age at exposure (for instance, current age of the respondents at the time of the interview minus 15 years for the long-term exposure). Then maybe one potential interpretation of the results is that exposure before birth, during early childhood and childhood (age -2 to age 9, obtained by subtracting 15 years from age 13 to 24) matters more than exposure during adolescence (age 8 to 19, obtained by subtracting 5 years from age 13 to age 24) for increasing the odds of being subjected to interpersonal violence.

Thank you for raising this interesting issue. First, we would like to clarify that we do not use repeated cross-sections. Instead, we pool nine cross-sectional VACS datasets from nine countries, resulting in a single nine-country cross-sectional dataset collected at various points in time. Given that the political violence events occurred in different locations at different points in time and that the respondents were quite heterogeneous in age (13-24) within each location, we unintentionally observe children, adolescents and young adults who experienced political violence at very different ages for any given location.

In fact, as discussed in reply to your previous comment, we aggregate the total number of political violence events per administrative area (weighted by local population size) over overlapping 1, 3, 5 and 15-year periods for four separate regressions. We did so as we expected events from a recent period to have stronger associations with VAC than events

measured over a longer period. Given that a location may have fixed effects which may lead to a higher tendency to have political violence, we decided against including multiple distinct PolV periods simultaneously, as such PolV variables would suffer from multicollinearity.

Interestingly, the 15-year period turned out to be most associated with VAC, even though some political violence events may have happened a long time ago. This suggests the importance of structural changes in the economy and/or society because of there being more events. If we understand your comment correctly, what you say would be true (and very interesting!) if PolV was spread across time equally.

However, the patterns of PolV are quite variable across time and space, suggesting randomness from the point of view of the respondents (that is, at the micro-level). Having said that, even if PolV may essentially be random for individuals (and especially adolescents and young adults!), we still do not interpret our findings as causal.

4. Political violence may cause migration, which may potentially lead to endogenous sample selection in the sample and selection bias in the estimates. If children with a higher risk of interpersonal violence are more likely to stay in the provinces with high political violence, and children with lower risk are more likely to move to the low-violence regions, this may lead to upward bias in the estimates. Do the VACS surveys provide any information on migration?

Thank you for raising the possibility of migration-related selection bias, which is an important point.

Forced displacement induced by political violence could bias our estimates, likely toward understating the true effect (that is our estimates are conservative). We would need more accurate (current and past) location data about the VACS respondents to test this formally. However, such data is not available for good reasons - to protect the respondents.

Please note that many places we study are not intense war zones. In fact, VACS are not typically conducted in crisis settings (at least not at the time when we accessed the VACS data). We do not study war as such but political violence, which is quite prevalent, even in countries often considered peaceful like Namibia (which is in our sample). The raised concern could have been larger had we studied more intense crisis settings.

In the VACS, only Namibia asks a migration-related question, namely "Have you ever moved to another region in your country?". The follow-up question on the reason for moving has no answer option for conflict-induced displacement. Upon assessing the potential association between the migration indicator and VAC outcomes, we found no significant association ($p > 0.40$).

In addition, we examined national displacement patterns for potential selection using IDMC displacement data for each country (<https://www.internal-displacement.org/>). Seven of the nine countries do not have conflict-related displacement at least in the six years preceding our interpersonal violence-reporting window. The only recent displacement occurred in Nigeria and Mozambique, and in both cases the VACS sampling frame had already excluded the most insecure, displacement-affected clusters to ensure the safety and security of enumerators and respondents. In these countries, any misclassification would attenuate, not inflate, our estimates (that is our estimates are conservative).

Finally, we would like to point out that our approach involves matching ACLED data and VACS data by quite large administrative areas. As we say in the Methods section: “Across the nine countries we study, there are 723 sub-national administrative units which range in size from 8.60 to 84,750 square kilometres (mean=11,073; s.d.=17,744), with populations ranging from 14,460 to 6,321,017 (mean=694,333; s.d.=1,162,791). The mean number of VACS interviews per administrative unit is 49.01 (s.d.=114.9).” It is hence likely that internal displacement may have occurred within the same administrative area (such as provinces in Cote d’Ivoire or local government authorities in Nigeria). Had we been able to narrow down the location of the respondents in the VACS more accurately, then this displacement bias might have been larger. As it is, we believe that this is not likely to shape our findings significantly.

5. The measure of long-term exposure (past 15 years) includes the measure of short-term exposure (past 5 years). Is there a way to differentiate between the two? For example, the authors could include in the same regression exposure in the past 5 years (t-1 to t-5) and exposure in the 10 preceding years (t-6 to t-15).

Thank you for the suggestion, which we have now estimated as a robustness check and included on page 8. Following your suggestion, we re-estimated our main result using two non-overlapping windows of exposure to past political violence of 1–5 years and 6–15 years before the survey reference period and included both in the same logistic model. Collinearity is modest, so the two coefficients are well identified. The estimates in Table A9 in the Supplementary File indicate that political violence exposure in the preceding 6–15 years increases EV, AV, and MV, although the effect is statistically significant only for EV. This confirms that it is past political violence from many years (more than five) that is associated with (emotional) violence against adolescents and young adults today.

6. I don’t think the authors provide a compelling explanation for why only the long-term exposure matters and the short-term exposure does not. Could the authors elaborate more on possible explanations, without indulging in speculation?

Thank you for inviting us to explain our results in more detail. Our core finding is that the history of political violence of the last fifteen years matters for the current VAC status of adolescents and young adults. We expanded our discussion of possible mechanisms,

drawing on the related literature of the role of political violence for intimate partner violence.

We now propose five mechanisms that may be at play and review each briefly. The postulated mechanisms are (i) culture of insecurity, (ii) social learning of violence, (iii) social norms, (iv) socio-economic hardship, and (v) the consequence of the breakdown in civil society and government service provision.

However, we cannot formally test the role of these mechanisms, given the structure of our data. In that, we are aligned with the literature on violence against children and intimate partner violence in post-conflict settings, which shows a lack of rigorous studies testing these mechanisms, especially across countries. We believe that these mechanisms in fact are not mutually exclusive but may re-enforce each other.

Finally, we believe that our findings are broadly consistent with localised case studies from settings with political violence, which we review in the discussion as well. We hope that these changes do justice to your request for more possible explanations, without indulging in speculation.

Reviewer #3

REVIEW OF PAPER

The title of the reviewed paper is 'Past political violence and interpersonal violence against children and youth: Micro-level evidence from nine African countries'.

Summary

This paper presents the findings from a groundbreaking study that examines the relationship between past political violence and subsequent interpersonal violence against children and youth. The authors investigate how exposure to political violence may increase the risk of various forms of interpersonal violence, focusing specifically on children and youth. This analysis utilizes nationally representative data from Côte d'Ivoire, Kenya, Malawi, Mozambique, Namibia, Nigeria, Uganda, Zambia, and Zimbabwe. The study combines multi-country, cross-sectional microdata from the Violence against Children and Youth Surveys (VACS) with conflict event data from the Armed Conflict Location and Event Data (ACLED), using geographic and temporal identifiers for correlation. Logistic regression analyses are employed to assess the link between prior exposure to political violence and the increased likelihood of children and youth experiencing physical, emotional, or sexual violence. First, the study explores whether short-term or long-term exposure to political violence heightens the chances of children and youth encountering recent incidents of physical, sexual, or emotional abuse. The authors also investigate whether the adverse effects of exposure to political violence differ across various categories of such violence. These findings underscore the necessity of considering the political and historical context when devising strategies to prevent and address violence against children and youth, whether in families, schools, or early intimate relationships. The results pave the way for future research and initiatives aimed at achieving Sustainable Development Goal 16.2, which focuses on the prevention of violence against children and youth.

Methodology

The study used a quantitative, cross-sectional, and observational design to analyze secondary data. The Violence Against Children Surveys (VACS) were conducted at a single point in time, which differentiates them from longitudinal studies. Researchers observed and analyzed existing data, focusing on variables like experiences of violence and exposure to conflict, using datasets from VACS and the Armed Conflict Location & Event Data Project (ACLED) rather than collecting primary data. All Violent Against Children and Youth Surveys (VACS) from countries in the Armed Conflict Location and Event Data (ACLED) database were included and matched with ACLED data from the 15 years preceding each VACS. The resulting merged dataset includes VACS collected between 2013 and 2019, covering 35,437 individuals aged 13 to 24 from nine countries: Côte d'Ivoire, Kenya, Malawi, Mozambique, Namibia, Nigeria, Uganda, Zambia, and Zimbabwe. Together, these VACS data represent approximately 30% of Africa's population. The Violent Against Children and Youth Surveys (VACS) are nationally representative household surveys that focus on children and youth aged 13 to 24 years. The Armed Conflict Location and Event Data (ACLED) codes conflict events across countries, using information gathered from publicly available news sources, along with temporal and precise geographic identifiers.

Thank you for this succinct summary of our paper.

Strengths of the Paper

- *In citing other works as evidence or presenting factual information, this paper acknowledges the authors and studies referenced.*
- *Methodology and Discussion were thoroughly explained.*
- *Sufficient Use of Visuals (Tables, Graphs, and Charts) made the data easy to interpret with visual representation.*
- *The authors identified and stated several important topics that need further research related to the subject matter.*
- *The results were rigorously tested and subjected to a placebo test to verify their accuracy and reliability.*
- *The secondary data (VACS and ACLED) used is valid and reliable.*

Thank you for this feedback - we agree.

Shortcomings of the Paper

- *In the Introduction, the terms in the topic are not properly expanded with case studies and scenarios.*

We revised our introduction and provided a range of references to studies to contextualise our paper. Key terms like “violence against children, adolescents and young adults” and “political violence” are defined formally in the new sub-section on “Measures” in the revised section on “Methods”.

- *The authors do not explain the chosen methodology or its limitations.*

We respectfully disagree. We offer a revised section on “Methods” placed towards the end of the paper (in line with editorial guidelines) and we discuss several limitations towards the end of the “Discussion” section on pages 11-12.

- *The paper does not provide any recommendations for resolving these issues.*

We respectfully disagree. We offer a discussion of policy implications to help achieve SDG 16.2 (“End abuse, exploitation, trafficking and all forms of violence against and torture of children”) at the end of the “Discussion” section. In that discussion, we also suggest model interventions which can help achieve a reduction of VAC in settings shaped by past political violence.

- *The author does not outline the limitations and misconceptions in research concerning the subject.*

We respectfully disagree. In the “Introduction”, we discuss the limitations in the literature on violence against children, adolescents and young adults, indicating the need for further research on the links between political violence and VAC. We also discuss several

limitations in our empirical approach towards the end of the “Discussion” section on pages 11-12.

- *The data is cross-sectional and, as such, cannot establish causality; it merely highlights correlations. While exposure to conflict is associated with an increased risk of violence, one cannot conclude that one directly causes the other.*

We completely agree with this statement! Methodologically, we would have liked to establish causality, but it is not possible to experimentally assign levels of political violence, nor to identify a robust instrument that could aid in establishing causality. We double checked our manuscript to ensure that we only speak of correlations or associations.

- *The paper does not explore the various fields that relate to or influence the topic.*

We respectfully disagree. Our work is relevant for and draws on global public health, gender studies, political science, peace research and development studies. We cite papers from across these literatures and expect to be cited by them in turn, once our paper will be published. As evidence of our engagement with interdisciplinary research, we refer to our updated list of references.

- *A layperson may find it difficult to understand the paper, particularly the results and discussion sections.*

In a sense we agree with this comment but note that our paper is not intended for a lay audience. We aim to inform researchers working in the above-mentioned fields as well as experts working in child protection and public health. We believe that our paper is well written and easily accessible to these audiences. We also double checked our abstract to ensure that this is accessible to a wider audience.

Recommendations

- *The limitations in causality, potential biases, and lack of qualitative depth must be acknowledged when interpreting results.*

We agree with this recommendation and have edited the manuscript to ensure that we do not make any claims of causality and that we discuss potential sources of bias.

- *Detail how the findings can assist policymakers, practitioners, or local communities in advancing Sustainable Development Goal 16.2.*

We agree that it is important to suggest how our findings can assist policymakers, practitioners, or local communities in advancing Sustainable Development Goal 16.2. We do so by discussing the implications of our finding in the revised “Discussion” section. Specifically, we say on page 12:

“There are examples of recent parenting programmes to reduce domestic violence that could be adapted and considered for implementation in contexts of political violence (49). There is a need to develop further school-based programming for conflict-affected areas, and there is little evidence of effectiveness of any school-based violence prevention programmes in any area with past political violence (50). Programmes to prevent intimate partner violence may hold special promise, as these programmes can often simultaneously reduce IPV and VAC within the same households (51), and examples exist of interventions which can be deployed in conflict and post-conflict contexts (29, 52).”

Reviewer #4
